# The Enteric Glia and Its Modulation by the Endocannabinoid System, a New Target for Cannabinoid-Based Nutraceuticals?

**DOI:** 10.3390/molecules27196773

**Published:** 2022-10-10

**Authors:** Laura López-Gómez, Agata Szymaszkiewicz, Marta Zielińska, Raquel Abalo

**Affiliations:** 1High Performance Research Group in Physiopathology and Pharmacology of the Digestive System (NeuGut), Department of Basic Health Sciences, University Rey Juan Carlos, 28922 Alcorcón, Spain; 2Department of Biochemistry, Faculty of Medicine, Medical University of Lodz, Molecolab, 92-215 Lodz, Poland; 3R & D & I Unit Associated with the Institute of Medicinal Chemistry (IQM), Spanish National Research Council (CSIC), 28006 Madrid, Spain; 4Spanish Pain Society Working Group on Basic Sciences in Pain and Analgesia, 28046 Madrid, Spain; 5Spanish Pain Society Working Group on Cannabinoids, 28046 Madrid, Spain

**Keywords:** cannabidiol, endocannabinoid system, enteric glial cells, enteric nervous system, gastrointestinal system, nutraceuticals, palmitoylethanolamide

## Abstract

The enteric nervous system (ENS) is a part of the autonomic nervous system that intrinsically innervates the gastrointestinal (GI) tract. Whereas enteric neurons have been deeply studied, the enteric glial cells (EGCs) have received less attention. However, these are immune-competent cells that contribute to the maintenance of the GI tract homeostasis through supporting epithelial integrity, providing neuroprotection, and influencing the GI motor function and sensation. The endogenous cannabinoid system (ECS) includes endogenous classical cannabinoids (anandamide, 2-arachidonoylglycerol), cannabinoid-like ligands (oleoylethanolamide (OEA) and palmitoylethanolamide (PEA)), enzymes involved in their metabolism (FAAH, MAGL, COX-2) and classical (CB1 and CB2) and non-classical (TRPV1, GPR55, PPAR) receptors. The ECS participates in many processes crucial for the proper functioning of the GI tract, in which the EGCs are involved. Thus, the modulation of the EGCs through the ECS might be beneficial to treat some dysfunctions of the GI tract. This review explores the role of EGCs and ECS on the GI tract functions and dysfunctions, and the current knowledge about how EGCs may be modulated by the ECS components, as possible new targets for cannabinoids and cannabinoid-like molecules, particularly those with potential nutraceutical use.

## 1. Introduction

The digestive system is the primary site of energy and nutrient absorption and plays a key role in metabolic homeostasis, i.e., “the capacity of organisms to maintain stable conditions on its composition and properties by compensating changes in their internal environment through the regulated exchange of matter and energy”. [1]. Within the gut wall lies the largest endocrine and immune system of the body, as well as the enteric nervous system (ENS) [2]. The gastrointestinal (GI) tract is connected with the central nervous system (CNS), through the extrinsic innervation of the autonomic nervous system (ANS) and stress hormones. Thus, the existence of an important brain-gut axis has been recognized [3].

Whereas the neurons in the ENS have been widely studied throughout time, the enteric glial cells (EGCs) have received less attention [4,5,6]. Numerous GI conditions have been found to be associated with alterations in the numbers and functions of these cells [4,7,8,9].

The term nutraceutical was first defined in 1989. This term is a combination of the words “nutrition” and “pharmaceutical” and refers to “food components or active ingredients present in food that have positive effects for well-being and health, including the prevention and treatment of diseases” [10]. The endogenous cannabinoid system (ECS) is a well-recognized modulator of the GI tract [11,12,13,14,15,16]. The components of the ECS are found in many cell types within the GI tract, including the ENS. Not surprisingly, exogenously administered cannabinoids have profound effects that may be beneficial for the treatment of some GI conditions [14,17,18,19], and adverse GI effects of their use have also been recognized (i.e., cannabinoid hyperemesis [20,21] and small bowel intussusception, [22]). However, the relationship between the components of the ECS and EGCs and the mechanism by which nutraceuticals may act through them have not been well established yet.

In this review, we first describe the ENS and its functions, with a particular focus on EGC physiopathology. Thereafter, we review the important role of the ECS in the GI tract and ENS. Finally, we explore the interaction between the ECS and EGCs and the effects of natural cannabinoids and cannabinoid-like molecules with potential nutraceutical use in GI disorders.

Although further studies are needed to define the connections between the ECS and EGCs as a possible target to treat or reduce alterations associated with GI disorders, the use of cannabinoids may be beneficial in prevalent pathologies such as inflammatory bowel disease (IBD) and, maybe, other types of GI pathologies displaying ENS inflammation. The information included in this paper could serve as a starting point for future research.

## 2. The Enteric Nervous System

The ENS constitutes a complex network of neurons and accompanying glial cells that control the major functions of the GI tract [23]. In detail, the ENS is composed of intrinsic sensory neurons (intrinsic primary afferent neurons, IPANs), excitatory and inhibitory interneurons, and motor neurons. The complexity of the ENS contributes to the independency of its action: sensory neurons receive external inputs, then interneurons integrate the signals, and together with motor neurons generate outputs. Moreover, ENS may receive and process the signals from the CNS [24].

Within the ENS, neuronal and glial cells are organized in myenteric and submucosal plexuses. The first one is located between the two layers of smooth muscle (circular and longitudinal muscle layer) and is involved in the coordination of GI motility, while neurons of the submucosal plexus (located between the mucosa and the muscle layers) participate in secretion and absorption of water and electrolytes [2].

### 2.1. Enteric Neurons

IPANs possess mechano- or chemosensory activity, and besides the straight signal reception, they are able to receive and process the message of the intensity, duration, and pattern of stimuli. These neurons usually form a circumferential internetwork encircling the intestine. Within the group of IPANs, several classes may be listed, for example, according to their localization (myenteric/submucosal plexus) or the direction of signal transduction. Therefore, IPANs can receive, integrate and reinforce signals both locally and across the network (alike interneurons) [25,26].

Interneurons, like IPANs, may be divided into ascending or descending. Furthermore, within the population of interneurons there are several classes that may be distinguished neurochemically and the proportion of interneurons in these classes may differ between the parts of the GI tract, that may reflect the regional diversity in the motor patterns in the intestines [4,27,28].

The last group of neurons in the ENS are motor neurons, which are divided into two subgroups: inhibitory and excitatory. They participate in the control of intestinal motility as they contribute to the contractions and relaxations of the circular and longitudinal smooth muscles in a mechanism dependent on acetylcholine (ACh) (excitatory neurons), or nitric oxide (NO), vasoactive intestinal peptide (VIP) and pituitary adenylate cyclase-activating polypeptide (PACAP) (inhibitory neurons) [2].

### 2.2. Enteric Glial Cells

Glial cells located in the GI tract are also known as enteric glial cells (EGCs). At first, they were simply considered as structural support for the ENS. It is now well recognized that they participate in several processes crucial for the GI tract [29,30].

Hanani et al. [31] classified EGCs into 4 subgroups based on their morphology. Type I EGCs, named “protoplasmic”, are star-shaped cells with short, irregularly branched processes, resembling protoplasmic astrocytes in the CNS. Type II (fibrous) EGCs are elongated glia with interganglionic fiber tracts. Type III (mucosal) EGCs possess long-branched processes. Finally, type IV (intermuscular) EGCs are the elongated glia accompanying the nerve fibers and encircling the smooth muscles.

EGCs may also be subgrouped according to the molecular or functional differences due to the heterogeneities in receptors and channels expressed on their surface. In particular, several proteins are often used to identify EGCs, i.e., calcium-binding protein S100 [9,32], glial fibrillary acidic protein (GFAP) [9,33] and the transcription factors: SOX-8, SOX-9, SOX-10 [34] (Figure 1). Interestingly, Hanani et al. [35] and others [36], showed that EGCs are interconnected and electrically coupled by gap junctions and form an extensive functional glial network [37].

EGCs play a role in intercellular communication, intestinal barrier formation and support, as well as control of the GI motility, immune response, and visceral sensitivity (Table 1).

#### 2.2.1. EGCs and Intercellular Communication

EGCs are generally considered as non-excitable cells because they do not generate action potentials [73]. However, they communicate through calcium (Ca^2+^) dependent signaling (this is by far the ion that has been more clearly involved in EGCs signaling activity) and therefore they are able to integrate information transmitted by other glial cells, neurons, immune cells, and other cells [74]. The intercellular communication in EGCs (gliotransmission [75]) occurs mainly through the propagation of Ca^2+^ waves via connexin 43 (Cx43) hemichannels, through which adenosine triphosphate (ATP) or other molecules (even Ca^2+^) may be released to act on close glial, neural, or immune cells. Thus, Cx43 is necessary for intercellular communication between EGCs, and interfering with EGC communication through these hemichannels profoundly alters GI function [76]. Besides the above mechanism, EGCs may also communicate with adjacent cells through exocytosis.

In general, EGCs, similarly to astrocytes in the CNS, express receptors for neurotransmitters on their surface and are able to release neurotransmitters (gliotransmitters), therefore they participate in neuronal communication in the ENS [77,78,79,80]. According to in vitro studies, EGCs are susceptible to activation by intrinsic (from enteric neurons) or extrinsic (from autonomic or primary afferent neurons) neural pathways. ATP is a major neurotransmitter involved in extracellular signaling in the ENS and the regulation of GI motility, secretion, or synaptic transmission [81]. It was demonstrated that EGCs respond to ATP by increasing intracellular levels of Ca^2+^. Gulbransen et al. [82] characterized the cross-talk between subgroups of neurons and EGCs in the guinea pig colon, where intrinsic fibers did not stimulate the increase of glial Ca^2+^ levels, but the activation of EGCs was significantly reduced after the ablation of extrinsic innervations. The same effect was elicited by selective sympathetic denervation. These results indicate that EGCs discriminate the activity of specific neural pathways (sympathetic). More recently, optogenetic stimulation combined with calcium imaging allowed to demonstrate the spatial and functional formation of neuro-glia units in the ENS, in which enteric neurons transmit signals to EGCs through pannexin channels using paracrine purinergic pathways [83].

Besides receiving the signaling, EGCs may induce transmission, because they are able to release neurotransmitters (i.e., glutamine, glutamate) [84]. Furthermore, through the presence of gamma amino butyric acid (GABA) transporter, GAT2, in the EGCs, these cells participate in GABA resynthesis [85,86]. Moreover, EGCs may be involved in nitrergic neurotransmission, as they demonstrate immuno-reactivity to L-arginine, the NO precursor [87,88].

#### 2.2.2. EGCs and the Intestinal Barrier

Mucosal EGCs are involved in intestinal epithelium differentiation, migration, adhesion, and proliferation and they constitute a link between the epithelial cells and submucosa neurons [38].

Interestingly, mucosal EGCs form a structure named ‘neuropod’, which is a cellular process that contains hormones and connects with enteroendocrine cells, scattered throughout the intestinal epithelium. These enteroendocrine cells participate in gut chemosensing and, in response to changes in environmental conditions inside the gut, they release amines and peptides acting locally or peripherally. Transmitters produced by enteroendocrine cells influence the EGCs and on the opposite, enteroendocrine cells may be a target for enteric glia. According to Bohórquez et al. [39], the formation of neuropods is regulated by neurotrophic factors, such as nerve growth factor β (NGF-β) or artemin.

One of the important molecules involved in intestinal barrier support is glucagon-like peptide 2 (GLP-2). Noteworthy, the presence of the receptor for GLP-2 was confirmed in the EGCs. Although the involvement of EGCs in intestinal barrier integrity has not been proved so far, EGCs appear to be an important factor that supports this barrier [42]. Moreover, besides GLP-2, several molecules produced by EGCs have emerged to be involved in intestinal barrier formation, i.e., proepidermal growth factor (proEGF), transforming growth factor (TGF)-β, S-nitrosoglutathione or 15-deoxy-Δ12,14-prostaglandin J2 (15d-PGJ2) [41,42,43,44,45].

#### 2.2.3. EGCs and GI Motility

A loss of EGCs causes a disruption of GI motility. In particular, Aubé et al. [46] demonstrated that progressive loss of enteric glia in transgenic mice (mice expressing haemagglutinin (HA), that received activated HA-specific CD8^+^ T cells) resulted in prolonged GI transit. Nasser et al. [47] observed a reduced GI peristalsis in vivo (prolonged upper GI transit) and impaired motility in vitro (a decrease in basal tone and the amplitude of the contractility in response to electrical field stimulation) in mice that received fluorocitrate (selective gliotoxin). Likewise, other researchers confirmed the involvement of Cx43 in GI motility [75,76]. Interestingly, Rico et al. [48] suggested that human EGCs are involved in the control of GI motility through the coordination of sensory and motor signaling.

As already mentioned, ATP is an important molecule involved in signaling transmission in the ENS. ATP interacts with three types of purinergic receptors (P1 receptors, P2X receptors, and P2Y receptors) [81]. In the CNS, the presence of P2X7 receptors on the surface of myelin sheaths allows to detect ATP released from axons [49]. P2X7 receptors were also found on the intramuscular glia in the GI tract [79]. Noteworthy, Gulbransen et al. [89] confirmed that purinergic signaling may constitute a link between neuronal and glial cells in the ENS as EGCs isolated from the guinea pig colon respond to ATP in vitro.

#### 2.2.4. EGCs and Immune System Cross-Talk

The EGCs participate in inflammation and are the first line of defense against pathogens [50]. Bush et al. [51] generated transgenic mice through the ablation of GFAP-positive glial cells from the jejunum and ileum that resulted in fulminating and fatal jejuno-ileitis. The ablation of enteric glia caused a severe inflammation leading to the degeneration of neurons in the ENS and evoked hemorrhagic necrosis of the small intestine. Authors compared the micro- and macroscopic alterations within the GI tract to the pathology in the course of IBD in rodent models and in humans [51]. IBD is a chronic inflammation of the GI tract, with two major types, Crohn’s disease (CD) and ulcerative colitis (UC).

In addition to their homeostatic role in supporting a healthy barrier (mentioned above), EGCs, depending on the stimulation (e.g., inflammation or following injury), may be activated and switched into a reactive, pro-inflammatory phenotype (reactive EGCs) [52,53] (Figure 2). Reactive EGCs display increased ability to proliferate (i.e., experimentally induced colitis promoted EGC mitosis in the myenteric plexus [54]), increased c-fos expression, and a change in the expression of EGC markers or surface receptors [55]. Thus, following the incubation with interleukin-1β (IL-1β), there is an increase in the expression of NGF receptor TrkA [56], endothelin-1 receptor B (ET-B) [57], toll-like receptor (TLR) 4 [58], and bradykinin receptor 1 (BR1) [59]. TrkA receptor is up-regulated in response to lipopolysaccharide (LPS) stimulation [56]. Besides, the expression profile of major histocompatibility complexes (MHC) changes from predominance of MHC class I (MHC I) under physiological conditions to increased expression of MHC class II after exposure to enteroinvasive *Escherichia coli* in culture [60]. Interestingly, an increased expression of MHC II was found in CD patients compared with healthy controls [61,62]. In addition, the expression of GFAP may be induced by tumor necrosis factor-α (TNF-α), IL-1β, LPS, or LPS+interferon-γ (IFN-γ) [57,63,64]. In vivo, LPS-induced intestinal inflammation resulted in an increased GFAP expression in the rat myenteric plexus [65]. The challenge with LPS and IFN-γ increases the expression of S100-β [66]. Thus, EGCs recognize inflammatory stimuli and, once activated, produce, and secrete S100-β, activating inducible NO synthase (iNOS) and NO production. Using rectal and duodenal biopsies, this mechanism has been confirmed in both UC and celiac disease patients [65,67]. Moreover, reactive EGCs can release neurotrophins, growth factors, or cytokines and therefore EGCs recruit immune cells (macrophages, neutrophils, mast cells) into the colonic mucosa [68,69,70].

#### 2.2.5. EGCs and Visceral Sensitivity

A recent review has suggested that enteric glia may be a new player in abdominal pain occurrence [8]. Abdominal pain is a frequent symptom associated with both acute and chronic GI disorders. For example, it occurs during IBD flares or also during remission phases. In the case of irritable bowel syndrome (IBS), a disease associated with a disturbed brain-gut axis [3], visceral hypersensitivity is the main feature. This pathology is characterized by abdominal pain combined with impaired motility (either accelerated (diarrhea-predominant IBS, IBS-D), slowed down (constipation-predominant IBS, IBS-C), or mixed bowel habits, IBS-M).

EGCs regulate the key properties mediating the development of visceral hypersensitivity: neuronal sensitivity, firing patterns, and network activity in the periphery, brain, and spinal cord. Immune activation and neuroplasticity (neuroinflammation) are essential in the generation of chronic abdominal pain. Although the precise mechanisms need to be defined, EGCs may contribute to sensitizing or activating nociceptors, through direct and indirect mechanisms. Additionally, EGCs may also regulate nociceptor sensitization/activation through the removal of neuromodulators [8].

Direct sensitizing mechanisms would involve the release of neuromodulators (ATP, GABA, IL-1β, neurotrophins), whereas indirect mechanisms would include antigen presentation (through MHC I and II), leading to activation of T cells that will release cytokines, and regulation of other immune cells like mast cells and macrophages, leading to the release of histamine and some cytokines (i.e., TNF-α, IL-1 β) [8]. Along these lines, it has been recently reported that pro-inflammatory signals induce glial Cx43-dependent macrophage colony-stimulating factor (M-CSF) production through protein kinase C (PKC) and TNF-α converting enzyme (TACE), further supporting the key role of EGC interaction with macrophages in the regulation of visceral hypersensitivity during chronic inflammation [71].

Furthermore, in a recent report using a rat model of colitis induced by intracolonic 2,4-dinitrobenzene-sulfonic acid (DNBS), Lucarini et al. [72] have demonstrated the essential role of enteric glia in the development of visceral hypersensitivity associated with intestinal inflammation. In this work, the gliotoxin fluorocitrate reduced intestinal damage and visceral sensitivity. Remarkably, a single injection of fluorocitrate reduced the colitis-induced overexpression of S100-β and transient receptor potential channel of subfamily V member 1 (TRPV1, a key component in nociception) not only in the colonic ENS but also in the dorsal root ganglia (DRG), satellite glia and astrocytes of the periaqueductal gray (PAG). Since fluorocitrate is unable to cross the blood-brain barrier, the neuroplastic effects in the PAG are unlikely due to direct inhibition of the central glia. Thus, EGCs have a pro-inflammatory and pro-nociceptive role during colitis, which is suggested to occur through the selective recruitment of mast cells and activated macrophages in the colonic submucosa and the occurrence of neuroplastic changes within the enteric circuits and along the pain signaling pathway, i.e., throughout the gut-brain axis [72].

#### 2.2.6. EGCs and Altered GI Functions

Importantly, local, or systemic conditions affecting the GI tract function have been described to be associated with changes in EGC numbers, expression profile, and functionality (Table 2), suggesting that EGCs may be an interesting target for treatment and/or prevention of those pathologies [90].

## 3. The Endocannabinoid System

The term cannabinoid comprises a group of at least 66 biologically active terpenophenols which are found in cannabis (*Cannabis sativa*) and their synthetic analogs [127]. Cannabinoids are molecules that act on the endogenous cannabinoid system (ECS), also known as the endocannabinoid system, and are usually divided into three main groups: phytocannabinoids (cannabinoids found in plants), endocannabinoids (endogenous compounds found in animals that modulate cannabinoid receptors); and synthetic cannabinoids (synthetic compounds that may or may not be structurally related that also produce agonistic effects in cannabinoid receptors) [128]. Figure 3 shows the molecular structure of the two cannabinoid compounds that have been more deeply studied in relation to EGCs.

The ECS is composed of cannabinoid receptors (CB1, CB2), their endogenous ligands (endocannabinoids, ECBs), and the enzymes involved in the biosynthesis and degradation of cannabinoids.

Cannabinoid receptors belong to the G-protein coupled receptors (GPCR) family. Their activation results in the inhibition of adenyl cyclase activity and suppression of voltage gated Ca^2+^ channels [129]. Noteworthy, CB receptors possess more than one endogenous agonist: anandamide (*N*-arachidonoyl ethanolamine, AEA) and 2-arachidonoyl glycerol (2-AG). ECBs are derivatives of the arachidonic acid, synthesized on demand from the membrane phospholipids in the post-synaptic cells in response to increased levels of intracellular Ca^2+^, released immediately after synthesis, and then diffuse throughout the cellular membrane without being stored in vesicles. The action of ECBs is mediated through CB1 or CB2 receptors. Noteworthy, ECBs exhibit different selectivity and affinity: AEA is a partial agonist of CB1 with very weak activity at CB2 receptors, while 2-AG is characterized as a potent agonist of both receptors. Besides these compounds, there are other ECBs that remain less known: 2-arachidonyl glyceryl ether (2-AGE, a CB1 selective agonist) [130], N-arachidonoyl dopamine (NADA, a CB1 agonist) [131], and O-arachidonoyl ethanolamine (O-AEA or virodhamine, a partial CB1 agonist and full CB2 agonist) [132].

Interestingly, it was demonstrated that ECBs may interact with other receptors. For example, AEA binds to TRPV1 [133,134]. The effects of TRPV1 activation depend on the site of action: when AEA interacts with pre-synaptic TRPV1 it promotes glutamate release, while the activation of post-synaptic TRPV1 by AEA leads to the reduction of glutamate signaling, inhibition of 2-AG biosynthesis and the blockage of the retrograde action at CB1 receptors. The multi-target action of ECBs may be related to the co-expression of CB receptors and TRPV1 channels in neuronal and non-neuronal cells. It was assessed that TRPV1 are co-localized with CB1 or CB2 receptors in the primary sensory neurons of the DRG in rats [135,136,137], perivascular neurons [138], vagus nerve [139], and in the axons of neurons in the CNS [140,141,142]. Moreover, CB receptors are co-expressed with TRPV1 in the endothelial cells of the brain microvessels (both CB1, CB2) [143], in the endothelial cells from the rodent mesenteric arteries with cirrhosis (CB1) [144], dendritic cells [145], muscle cells (in both rodents and humans), [146], osteoclasts [147], keratinocytes [148], and melanocytes [149]. G protein-coupled receptor 55 (GPR55) is an orphan receptor that constitutes another, non-classical target, for ECBs. Lysophosphatidylinositol (LPI) was identified as the endogenous ligand for GPR55 [150]. However, AEA and O-AEA can activate these receptors [151,152]. Finally, besides TRPV1 and GPR55, ECBs exhibit binding affinity at peroxisome proliferator-activated receptors (PPAR): AEA, O-AEA, and 2-AGE bind PPARα, 2-AG binds to PPARβ/δ, while AEA, 2-AG and 2-AGE bind to PPARγ in vitro [152].

After the activation of CB receptors, the remaining ECBs are degraded in the process of hydrolysis or oxidization. The first enzyme discovered to be involved in ECBs degradation was named fatty acid amide hydrolase (FAAH). Its most preferred substrate was found to be AEA. A few years later, other enzymes were discovered, and their properties were characterized: monoacylglycerol lipase (MAGL), α, β-hydrolase-6 (ABHD6), and α, β-hydrolase-12 (ABHD12) [153,154,155]. The process of oxidation involves cyclooxygenase-2 (COX-2) and several lipooxygenases [155].

### 3.1. The Endocannabinoid System in the Gastrointestinal Tract

The ECS components are widely expressed in the GI tract. The presence of CB1 receptors, the most predominant in the intestines, was confirmed in the enteric neurons [156], myocytes [157], and epithelial cells [158]. It should be emphasized that intestinal CB1 receptors participate in epithelial regeneration [159] and therefore play a crucial role in the maintenance of the intestinal barrier integrity [158]. Activation of CB1 receptors influences GI motility as it evokes the relaxation of the longitudinal smooth muscles. This action is a combination of the neurogenic effect and direct impact on myocytes [157]. Noteworthy, mRNA expression of CB1 receptors in the mouse colon does not change significantly during inflammatory conditions, i.e., induced by dextran sulfate sodium (DSS) and LPS [160].

CB2 receptors are present in the GI tract, but their expression is lower in comparison to CB1 receptors [14,156]. Similarly to CB1 receptors, LPS- or DSS-induced colitis does not alter the mRNA expression of CB2 receptors, but these receptors do play an important role during intestinal inflammation. In particular, the activation of CB2 receptors with a selective agonist (JWH-133) attenuates chronic colitis in IL-10 deficient mice [161,162]. Furthermore, according to Storr et al. [163], CB2 mRNA is up-regulated in mice with 2,4,6-trinitrobenzene sulfonic acid (TNBS) induced intestinal inflammation.

In animal models of intestinal inflammation (induced by DNBS or TNBS), the level of AEA is increased in the mucosa, but not in the muscular layer of the colon [164]. Furthermore, AEA is up-regulated in the colonic biopsies collected from patients with UC. However, the expression of 2-AG is not altered in both, animal, and human, inflamed colonic samples [164].

According to Grill et al. [160], MAGL mRNA expression decreases in mouse intestines in LPS- and DSS- induced colitis. Furthermore, it was found that the inhibition of enzyme activity increases the level of 2-AG and thus improves TNBS-induced colitis in mice [165]. Interestingly, Wasilewski et al. [166] investigated the impact of inhibition of the activity of enzymes involved in the degradation of ECBs (FAAH, MAGL) on the colonic secretion stimulated with forskolin (cyclic adenosine monophosphate, cAMP-dependent secretagogue), veratridine (voltage-dependent sodium channel activator) or bethanechol (cholinergic receptor agonist, resistant to the action of cholinesterases). The inhibition of FAAH activity abolishes the pro-secretory effect induced by forskolin, which is mediated through CB1/CB2 receptors. On the other side, the anti-secretory action of MAGL inhibitors is reversed by the CB2 receptor antagonist AM-631, but not by AM-251 (CB1 receptor antagonist).

The ECS is also involved in the control of GI peristalsis and transit [14]. In particular, CB1 receptors were pointed out as involved in the inhibitory effect of cannabinoids, both, under control [167,168,169,170,171] and under inflammatory conditions [172]. IBS, or chemotherapy-induced dysmotility are associated with alterations in CB1 and CB2 receptors expression/activity, levels of ECBs or ECB biosynthesis/degradation.

Thus, the levels of 2-AG in plasma were higher in IBS-D patients, while oleoylethanolamide (OEA) and palmitoylethanolamide (PEA) were lower in comparison to healthy participants [173]. On the contrary, OEA concentration in plasma was higher in IBS-C patients as compared to controls. The AEA levels were similar in all IBS subtypes. Interestingly, the ECB turnover may be a key factor in the pathophysiology of IBS as the mRNA expression of FAAH in the colonic biopsies was significantly lower in IBS-C patients in comparison to healthy controls and the IBS-D group [173]. These results suggest that the impaired process of ECB degradation may lead to sustained activation of CB receptors and be the cause of prolonged intestinal transit. Furthermore, the symptoms of IBS have been related to the genetic polymorphism of genes encoding CB1 receptors and FAAH [174,175]. For example, a higher number of AAT triplets in the *CNR1* gene (10 or more triplets) was noted in IBS patients in comparison to healthy controls. Moreover, this polymorphism was reported in patients with higher scores on the abdominal pain/discomfort symptoms scale [174]. The polymorphism in *CNR1 rs*806378 (CC vs. CT/TT) correlated with the efficacy of colonic transit: patients with TT variant were characterized with the fastest colonic transit [175]. The polymorphism in genes encoding FAAH may be related to IBS: conversion of 385C to A leads to decreased expression of FAAH. The CA/AA polymorphism was more frequent in IBS-D and IBS-M patients. There was a strong association between the FAAH CA/AA and accelerated colonic transit in IBS-D patients [176].

With regards to chemotherapy-induced dysmotility, it was found that the non-selective synthetic cannabinoid agonist WIN 55,212-2 (WIN), at a low dose devoid of central effects, partially decreased diarrhea associated with 5-FU treatment in rats, despite not being able to improve gut inflammation [177], suggesting a direct effect on the myenteric plexus. In contrast, animals treated with vincristine displayed paralytic ileus that improved when treated previously with the CB1-selective antagonist AM251, suggesting that vincristine-induced GI motor reduction is associated with an increase in ECS activity, involving CB1 receptors [178]. This is consistent with findings of ECS activation in other paralytic ileus conditions, like that produced by LPS administration [179]. Finally, WIN was not able to prevent repeated cisplatin-induced pica or gastric dysmotility (indirect markers of nausea/vomiting in non-vomiting species [180,181]) in the rat [182,183]. These results are conflicting with the known empiric use of cannabinoids to prevent chemotherapy-induced nausea and vomiting but might be related to similar mechanisms occurring during the so-called cannabis hyperemesis syndrome, suffered by heavy cannabinoid consumers [184], at least those genetically susceptible [20].

Finally, a role for ECS has been demonstrated in abdominal pain [185]. Thus, in early preclinical studies using colorectal distension (CRD), both CB1 and CB2 receptors elicited analgesic effects under basal conditions and during inflammation-induced hyperalgesia [186,187]. Interestingly, an endogenous cannabinoid tone able to activate CB1 receptors was found in response to noxious CRD [188]. Moreover, TRPV1 channels were early demonstrated to be involved in the development of chemically induced visceral hypersensitivity to CRD [189] and acute mechanical colonic hyperalgesia without prior chemical challenge [190]. Furthermore, compared with control animals, AEA levels and TRPV1 expression increased whereas CB1 receptor expression decreased in DRG from rats submitted to water avoidance stress for 10 consecutive days (a paradigm well known to produce visceral hypersensitivity to CRD), and these changes were prevented by prior administration of the non-selective cannabinoid agonist WIN or the TRPV1 antagonist capsazepine [191]. More recently, it has been proposed that modulation of both FAAH and MAGL by dual inhibitors (i.e., increasing both AEA and 2-AG) might be a good strategy to control visceral pain [192]. Importantly, changes in the ECS in some cerebral areas, at least those affecting CB1 and TRPV1, may contribute to visceral hypersensitivity development at the supraspinal level [193].

Whatever the mechanism, the preferred cannabinoid-based strategies to counteract visceral pain and other altered functions of the GI tract are those not capable of exerting psychotropic effects, including peripherally-acting CB1 agonists [167], CB2 agonists (which regulate inflammation but lack psychotropic actions), inhibitors of ECB degradation or other compounds like the phytocannabinoid cannabidiol (CBD), whose amazing number of molecular targets and lack of psychoactivity constitute an attractive alternative in this field [18]. Many other compounds in hemp (the variety of *Cannabis sativa* with lower concentrations of the main psychoactive compound of marijuana, Δ^9^-tetrahydrocannabinol), including other phytocannabinoids, terpens, alkaloids, steroids, flavonoids, and lignans are being studied for their potential use as nutraceuticals in the context of the GI ailments [194].

Likewise, other cannabinoid-like compounds may produce beneficial effects in those GI conditions [18]. However, as shown in the following section, thus far only a few studies have addressed the effects on the EGCs of phytocannabinoids and other cannabinoid-like molecules with potential nutraceutical use.

### 3.2. EGCs and the ECS

Cannabinoid compounds, due to their antioxidant and anti-inflammatory activity, could help to modulate the inflammatory processes in which EGCs intervene. However, not much is known about the specific interaction between the ECS and EGCs, probably because these cells do not seem to express the typical cannabinoid CB1 and CB2 receptors. In fact, a study using mouse colonic myenteric plexus found no co-localization of the CB1 receptor with specific markers for these cells, S100-β or GFAP [195]. Other immunostainings did not prove the presence of CB1 or CB2 receptors in EGCs in the GI tract (pylorus, duodenum, ileum, and colon) of cats [196]. In a study carried out in dogs, the CB1 and CB2 receptors were not seen either in the EGCs from the myenteric plexus, although weak-to-moderate immunoreactivity was detected in both neurons and glial cells from a few submucosal ganglia [197]. However, in a recent study carried out in cryosections of the distal ileum of horses, EGCs showed immunoreactivity for the CB1 receptor and PPARα [198]. Thus, the expression of cannabinoid receptors by the EGCs may be species-dependent. Not much is known about the expression and role of CB1 and CB2 receptors in human EGCs. Despite the apparent lack of expression of CB receptors in the ENS in most species evaluated, CB agonists and antagonists might exert indirect actions on the EGCs. For example, it was shown that LPS increased the expression of c-fos in rat ileal EGCs and enteric neurons, and this was attenuated by CB2 agonists. Since enteric neurons do express CB2 receptors under inflammatory conditions, LPS activation of EGCs could be secondary to CB2-mediated neuronal activation [156].

TRPV1 is a non-selective cation channel activated by exogenous plant-derived vanilloid compounds as well as by endocannabinoids (namely, anandamide). In a study carried out by Yamamoto et al. [199], TRPV1-immunoreactive signals were detected in EGCs of the myenteric plexus of wild-type mice but not in TRPV1 knockout mice. Altered expression of GFAP at early postnatal time points in knock-out mice suggested that TRPV1 could be involved in enteric glia maturation. Furthermore, the addition of a TRPV1 antagonist to EGC cultures from wild-type mice myenteric plexus, decreased the expression ratio of GFAP to S100-β [199]. Thus, anandamide (and exogenous vanilloid compounds, like capsaicin), acting through TRPV1, may be a regulator of EGCs development and function, a hypothesis that still needs to be proved.

In contrast, EGCs express PPARα [196,197], a nuclear hormone receptor to which transcription-related ligands bind, and whose activation may induce anti-inflammatory and antinociceptive effects [58]. Since 2002, there has been accumulating evidence of the interaction of endocannabinoid compounds with PPARα, which may also be activated by other compounds similar to cannabinoids, phytocannabinoids or synthetic cannabinoids [200]. Since this receptor is expressed in EGCs, some investigations have been carried out to analyze the modulation of the pro-inflammatory activity of these cells by cannabinoid/cannabinoid-like compounds able to activate PPARα.

The AEA-like molecules OEA and PEA have an affinity for different receptors, including nuclear receptors such as PPARα, channels such as TRPV1, and membrane receptors such as GPR119 (OEA) and GPR55 (PEA). Of the two, PEA is the only one whose effects on EGCs have been reported [58]. PEA exerts dose-dependent anti-inflammatory effects on mouse models of UC and in human biopsies. This is due to its ability to reduce EGC activation, with the involvement of a cascade of PPARα activation, decreased expression of S100-β and TLR4, reduced expression of pro-inflammatory proteins such as iNOS, COX-2, and TNF-α, decreased NO production, reduced myeloperoxidase activity in mucosal neutrophils and reduced macrophage infiltration [58].

Interestingly, in a rat model of HIV-1-associated diarrhea induced by administration of HIV-1 tat, PEA reduced diarrhea through PPARα and consequent blockade of TLR4/NFκB activation, of colonic submucosal EGCs activation and overexpression of S100-β and iNOS [201].

Alzheimer’s disease (AD) is one of the most common neurodegenerative disorders characterized by functional digestive disturbances, including infrequent bowel movements, constipation, and defecatory disorder. The possible benefits of PEA administration in counteracting enteric inflammation have been investigated in an AD mouse model (SAMP8; Senescence-Accelerated Mouse-prone 8) [202]. In addition, the effects of PEA on EGCs were tested in cultured cells treated with LPS and β-amyloid 1–42 (Aβ) to mimic AD conditions. In this study, SAMP8 mice showed an increase in the expression of S100-β in colonic tissues, suggesting the presence of reactive gliotic processes. PEA administration induced a reduction of enteroglial-derived S100-β protein expression indicating that PEA is able to blunt the gliotic reaction. In cultures, EGCs treatment with PEA counteracted the increment of S100-β, TLR-4, NF-κB p65, and IL-1β release induced by LPS and Aβ. These results suggest that the anti-inflammatory effect of PEA prevents the enteric glial hyperactivation and counteracts the onset and progression of the colonic inflammatory condition by selectively targeting the S100-β/TLR-4 axis on ECGs, resulting in an inhibition of the NF-κB p65 pathway and cytokines release [202].

To our knowledge, no report has been published evaluating the involvement of other components of the endocannabinoid system (degradation enzymes FAAH and MAGL, GPR55, GPR119, …) in the physiopathology of EGCs.

### 3.3. Nutraceuticals Acting on the ECS with Potential Effects on EGCs

Taking the previous information into account it seems that many cannabinoids and cannabinoid-like molecules might exert an action on EGCs, by modulating the different receptors or targets already mentioned.

Indeed, hemp and its derivatives can be a source of nutraceuticals enriched in bioactive compounds. In this context, different compounds found in hemp may be an interesting choice to regulate EGCs functions: Δ9-tetrahydrocannabinol is a CB2 partial agonist and PPARγ agonist (although this compound appears in hemp in low quantities, only 0.3% or less); CBD is a PPARγ agonist, TRPV1 agonist, and CB2 partial agonist; cannabigerol acts as a weak CB2 partial agonist and TRPV1 agonist; cannabichromene is an agonist for CB2 and TRPV1; and both cannabidivarin and cannabidiolic acid are TRPV1 agonists [18].

CBD is the only phytocannabinoid whose effects on EGCs have been reported. CBD is an interesting phytocannabinoid devoid of psychoactivity, with great potential for the treatment of different GI disorders [18] and appears to be a key regulator of glia-mediated neuroinflammation in the GI tract. Although CBD does not directly interact with CB1 or CB2 receptors, it indirectly interacts with other cannabinoid receptors. A study performed by De Filippis [50] demonstrated that the activity of CBD is, at least partly, mediated via the selective PPAR-γ receptor pathway because GW9662, a potent PPAR-γ antagonist, reversed CBD effects on S100-β production. Thus, in a mouse model of intestinal inflammation induced by LPS, CBD prevented the hyperactivation of glial cells, decreased the expression of S100-β and reduced the infiltration of immune cells (mainly mast cells) [50]. Accordingly, in human rectal biopsies from patients with UC biopsies cultured with CBD, there was a reduction of S100-β expression, preventing EGCs activation. The exact cellular signaling pathways responsible for the effect of CBD still remain unclear but PPAR-γ may act as a key receptor in its action during gut inflammation.

Likewise, the endocannabionid-like lipids OEA (derived from oleic acid) and PEA (present in many plant and animal food sources, like milk, tomatoes, soybean, or peanuts) may modulate these cells via PPARα activation.

Thus, different compounds capable of activating or inactivating the ECS components related to the EGCs (TRPV1, PPARα, PPARγ, CB2), with potential nutraceutical use, might exert interesting modulatory effects in different GI conditions, and should be specifically evaluated in these regards (Table 3).

## 4. Conclusions

Enteric glia have recently attracted attention as an important functional component of the ENS that may contribute to the development and maintenance of GI dysfunctions of local and systemic origin [6]. These dysfunctions are also associated with important alterations in the expression and function of the different components of the ECS, suggesting some important connections between this system and the enteric glia (Figure 4).

Thus, AEA and other ligands of TRPV1 could be important for EGCs maturation. Furthermore, cannabinoid and cannabinoid-like compounds can regulate EGCs activity, directly through PPARα receptors (PEA) and indirectly through PPARγ or CB2 receptors (under GI inflammatory conditions), exerting an anti-inflammatory effect that can be beneficial in IBD and, maybe, other types of GI pathologies displaying ENS inflammation (i.e., plexitis [252]). However, the studies available so far are strikingly scarce.

We hope further studies are performed in the near future to more precisely define the connections between enteric glia and the endocannabinoid system as a possible target to treat or prevent the different disorders that affect the GI tract and the brain-gut axis, and the possible usefulness of nutraceuticals like those proposed in this review (Table 3).

## Figures and Tables

**Figure 1 molecules-27-06773-f001:**
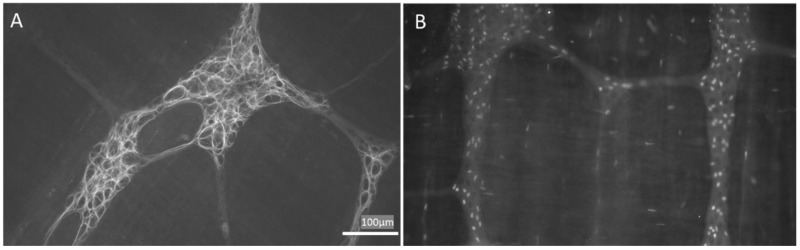
Appearance of enteric glial cells (EGCs). (**A**,**B**) are images obtained from the myenteric plexus of the rat distal colon; immunoreactivity to GFAP (**A**) and Sox-10 (**B**) are characteristic of EGCs. GFAP: glial fibrillary acidic protein. Images obtained by L.L.-G. (NeuGut-URJC).

**Figure 2 molecules-27-06773-f002:**
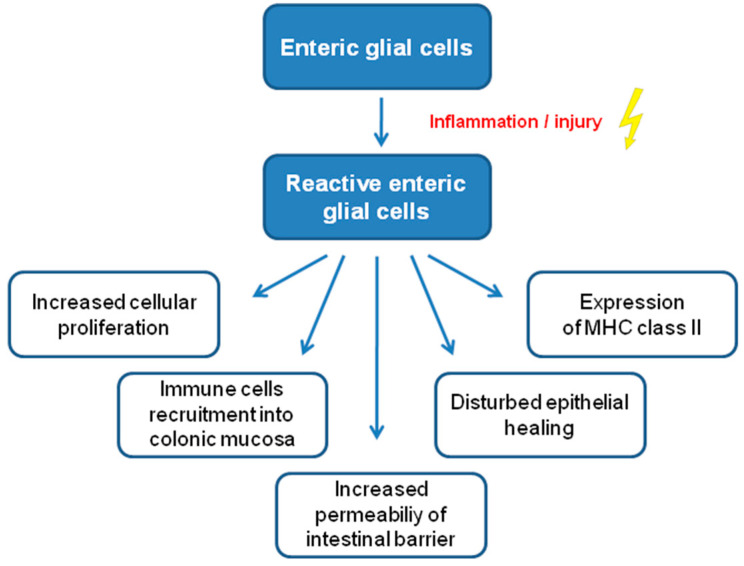
Summary of the properties of reactive enteric glial cells. Abbreviations: MHC, major histocompatibility complex.

**Figure 3 molecules-27-06773-f003:**
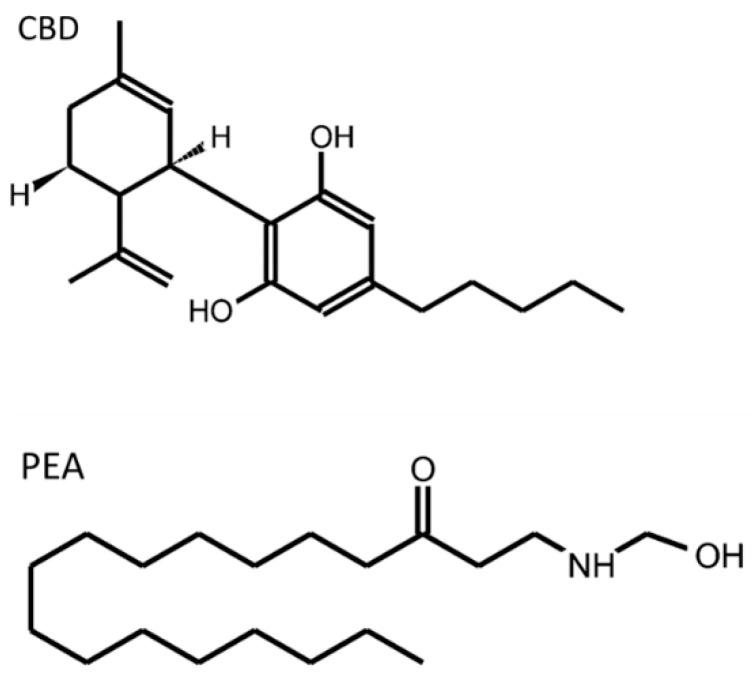
Chemical structure of cannabidiol (CBD) and palmitoylethanolamide (PEA). Molecules were drawn using http://biomodel.uah.es/en/DIY/JSME/draw.es.htm. (accessed on 1 September 2022).

**Figure 4 molecules-27-06773-f004:**
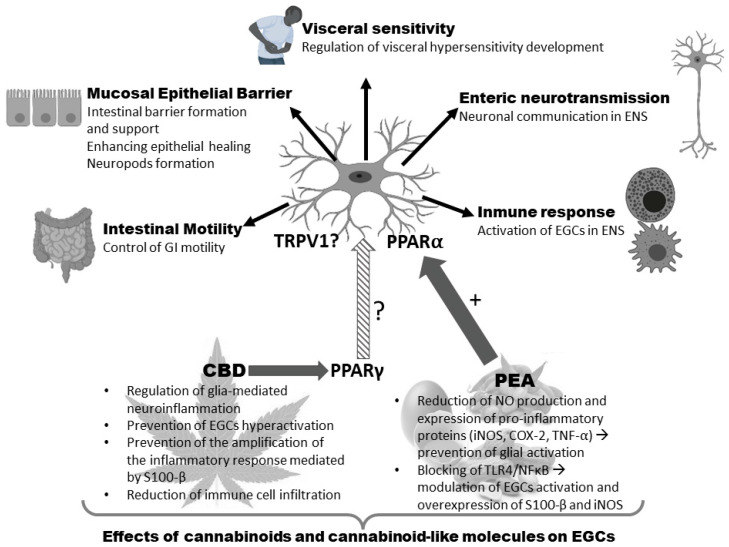
Effects of cannabinoids and cannabinoid-like molecules on enteric glial cells. +: activation; ?: indirect interaction through PPARγ. Abbreviations: CBD, cannabidiol; COX, cyclooxygenase; EGC, enteric glial cells; NFkB, nuclear factor-κB; iNos, inducible nitric oxide synthase; PEA, palmitoylethanolamide; PPAR, peroxisome proliferator-activated receptors; TNF, tumor necrosis factor; TLR4, toll-like receptor 4; TRPV, transient receptor potential vanilloid. Created with Biorender free application.

**Table 1 molecules-27-06773-t001:** Functions of the enteric glial cells in the gastrointestinal tract.

Aspect	Function	Localization	Mediators	References
Epithelial barrier	Intestinal barrier formation and support Enhancing epithelial healing Neuropods formation	Mucosa	proEGF TGF-β S-nitrosoglutathione 15d-PGJ2 NGF-β * Artemin *	[38,39,40,41,42,43,44,45]
Intestinal motility	Control of GI motility ^#^	Myenteric plexus	ATP	[46,47,48]
Enteric neurotransmission	Neuronal communication	ENS	ATP NFG GSH	[49]
Immune response	Activation of EGCs	ENS	MHC II class IL-1β IL-6 TGF-β proEGF GSH PGE2	[50,51,52,53,54,55,56,57,58,59,60,61,62,63,64,65,66,67,68,69,70]
Visceral sensitivity	Sensitizing/activating nociceptors	ENS	ATP GABA IL-1β neurotrophins	[8,71,72]

* Mediators released by enteroendocrine cells; ^#^ EGC loss results in impaired GI motility. Abbreviations: 15d-PGJ2, 15-deoxy-Δ12,14-prostaglandin J2; ATP, adenosine triphosphate; EGC, enteric glial cell; EGF, Epidermal growth factor; ENS, enteric nervous system; GABA, gamma amino butyric acid; GI, gastrointestinal; GSH, glutathione; IL, interleukin; MHC, major histocompatibility complex; NGF, nerve growth factor; PGE2, prostaglandin E2; proEGF, proepidermal growth factor; TGF, Transforming growth factor.

**Table 2 molecules-27-06773-t002:** Conditions affecting the gastrointestinal tract function for which a role of enteric glial cells has been described or suggested.

Condition	Species	Changes on EGCs	References
**PHYSIOLOGICAL**			
Aging	Rat	Loss of myenteric EGCs	[91]
Diet modification	Mice Rat	HFD reduces EGC density in duodenal mucosa and submucosal plexus Food restriction is detrimental to EGCs (but not neurons)	[92,93]
**GI DISEASES**			
IBD	Human	Glial markers (GFAP and S100-β) and GDNF are increased in inflamed areas of biopsies. In co-cultures, EGCs from CD patients increased intestinal permeability and epithelial proliferation	[55,94,95,96]
Celiac disease	Human	In EGCs from duodenal biopsies, increased S100-β and NO production	[67]
Chronic constipation	Human	Loss of ileal and colonic EGCs, in constipated patients, particularly in infectious megacolon (Chagas disease)	[97,98,99]
Postoperative ileus	Mice	In cultured EGCs, activation of IL-1R promotes an inflammatory response with increased IL-6 and MCP1 levels	[100]
IBS	Human Rat	Reduced immunoreactivity of S100-β in colonic biopsies (Independently from the IBS subtype) Exposure of EGCs to supernatants from mucosal human biopsies: IBS-C → reduced EGC→ proliferation; IBS-D and IBS-M→ impaired ATP-induced Ca^2+^ response of EGCs	[101,102]
PI-IBS associated with *C. difficile*	Rat	Exposure to toxin B produced cytotoxic and pro-apoptotic effects on cultured EGC	
Visceral hypersensitivity in IBS	Human Mice Rat	Increased expression of S-100, SP and TrkB (receptor for BDNF) in the colonic mucosa of IBS patients Increased expression of GFAP, SP and TrkB and induced VH in wild type but not BDNF+/− mice after administration of fecal supernatants from IBS-D patients. Butyrate enemas increased colocalization of GFAP and NGF in colonic EGCs, as well as NGF secretion.	[103,104,105]
Viral gastroenteritis	Human	EGCs stimulated with supernatants from ECCs infected with the human adenovirus 41 showed altered GFAP expression.	[106]
**SYSTEMIC DISEASES AFFECTING GI TRACT**			
Endotoxemia (systemic inflammation)	Rat	LPS systemically administered produced a dose-, time- and region-specific activation of EGCs (increased expression of S100-β and GFAP)	[107]
Obesity	Mice	In colonic whole-mount preparations, overexpression of S100-β (but not GFAP) and gliosis, with release of pro-inflammatory mediators. In cultured EGCs mimicking HFD-associated low-grade inflammation, increased SP and IL-1β production that may be related to dysmotility associated with obesity.	[108]
Diabetes	Mice Rat	Hyperglycemia promotes EGCs apoptosis involving Pdk1 and PI3K/Akt pathways. Lack of GDNF due to EGC loss, affects neuronal surviving, and GDNF supplementation limits neuronal loss.	[109,110,111]
Parkinson’s disease	Human	In colonic biopsies, increased expression of glial markers GFAP, S100-β, Sox10, accompanied by elevation of pro-inflammatory cytokines (TNF-α, IFN-γ, IL-1β, IL-6) at mRNA level. In colonic biopsies, GFAP over-expression.	[112,113]
Prion’s disease	Human	The spreading of pathological isoforms of cellular prion protein affects EGC in the GI tract.	[114,115,116]
HIV infection	Rat Mice	Intracolonic application of HIV1-tat protein produced lidocaine-sensitive S100-β and GFAP overexpression in submucosal plexus. Calcium signals from EGCs passed through Cx43 to glial cells of the spinal cord and the cerebral cortex, causing an inflammatory reaction, and cognitive loss. GI dysmotility and enhanced immune activation after treatment with HIV-1Tat + LPS, related to EGC release of IL-6, IL-1β and TNF-α and NF-κB activation; but not in glia from TLR4 KO mice	[117,118]
SARS-CoV-2 infection	Human	Enteric neurons and EGCs express ACE2 and TMRPSS2 and may be susceptible to invasion by the virus, this may lead to compromised immune response, cytokine storm facilitation, as well as alterations in intestinal motility.	[119]
**DRUG-INDUCED** **GI DISORDERS**			
Opioid-induced hyperalgesia and “narcotic bowel syndrome”	Mice	Upregulation of purinergic signaling in EGCs induced by prolonged opioid use and proinflammatory cytokine release, leading to gut barrier dysfunction and constipation	[8,120]
Cancer chemotherapy: oxaliplatin	Mice	In ileal whole-mount preparations, GFAP decreased in submucosal and myenteric plexus and S100-β increased in the myenteric plexus and mucosa. In distal colon, GFAP immunolabelling decreased whereas S100-β increased.	[121,122]
Cancer chemotherapy: 5-FU	Mice	Increased expression of S100-β protein in GFAP-positive cells during mucositis Pentamidine inhibits S100-β induced by 5-FU and this inhibits gliosis.	[123]
Cancer chemotherapy: irinotecan	Mice	Increased co-expression of GFAP and S100-β in irinotecan-treated tissues (duodenum, jejunum, ileum). Indirect relationship of mast cells with EGCs: forced mast cell degranulation, decreased the expression of GFAP and S100-β	[124]
Cancer chemotherapy: cisplatin	Mice	Chronic treatment with cisplatin reduces expression of S100-β, GFAP and SOX-10 in EGCs as well as that of ChAT and nNOS in myenteric neurons.	[125]
Cancer chemotherapy: others	Guinea pig	In cultured ECGs exposed to cytochalasin D (alters microfilaments), and nocodazole (alters microtubules), entry of calcium is reduced → other antineoplastic drug directed against elements of the cytoskeleton (taxanes, vinca alkaloids) might impair entry of calcium, and therefore alter EGC activity	[126]

Abbreviations: 5-FU, 5-fluorouracil; ACE2, angiotensin converting enzyme 2; ATP, adenosine triphosphate; BDNF, brain derived neurotrophic factor; CD, Crohn’s disease; ChAT, choline acetyltransferase; CNS, central nervous system; Cx43, connexin 43; ECC, enterochromaffin cell; EGC, enteric glial cell; ENS, enteric nervous system; GDNF, glial cell-derived neurotrophic factor; GFAP, glial fibrillary acidic protein; GI, gastrointestinal; HFD, high-fat diet; HIV, human immunodeficiency virus; IBD, inflammatory bowel disease; IBS, irritable bowel syndrome; IBS-C, irritable bowel syndrome with constipation; IBS-D, irritable bowel syndrome with diarrhea; IBS-M, mixed or alternating irritable bowel syndrome; ICC, interstitial cell of Cajal; IFN, interferon; IL-1R, interleukin 1 receptor; IL, interleukin; KO, knock-out; LPS, lipopolysaccharide; MCP1, monocyte chemoattractant protein-1; mRNA, messenger ribonucleic acid; NF-κB, nuclear factor kappa B; NGF, nerve growth factor; nNOS, neuronal nitric oxide synthase; NO, nitric oxide; Pdk1, pyruvate dehydrogenase lipoamide kinase isozyme 1; PI-IBS, post-infectious irritable bowel syndrome; PI3K/Akt, phosphatidylinositol 3-kinases/protein kinase B signaling pathway; SARS-CoV-2, severe acute respiratory syndrome coronavirus 2; SP, substance P; TLR, toll-like receptor; TMRPSS2, transmembrane protease serine 2; TNF, tumor necrosis factor; TrkB, tropomyosin receptor kinase B; UC, ulcerative colitis; VH, visceral hypersensitivity.

**Table 3 molecules-27-06773-t003:** Other nutraceuticals capable of activating or inactivating the ECS components related to the EGCs.

ECS Component	Nutraceutical (And its Natural Source)	Effect/Reference
CB2	*Harpagophytum procumbens* root extract	Activation [203]
	*β* *-caryophyllene (oregano, cinnamon, and black pepper)*	Agonist [204]
	*Olive oil*	Increase CB2 expression [205]
	*Lactobacillus fermentum MCC2760 **	Increase CB2 expression [206]
	*Lactobacillus acidophilus NCFM **	Decrease CB2 expression [207]
TRPV1	Capsaicin (chili peppers)	Agonist [208]
	Decursin (eggs)	Antagonist [209]
	Fish oil	Decrease TRPV1 expression [210]
	Omega 3 fatty acids	Activation [211]
	Probiotics: VSL#3	
	*Lactobacillus fermentum* CQPC03 *	Decrease TRPV1 expression [212]
	*Lactobacillus casei* Qian *	Decrease TRPV1 expression [213]
	*Lactobacillus reuteri* DSM 17938 *	Decrease TRPV1 expression [214] Antagonist [215]
PPAR α	Oleic acid	Agonist [216]
	Oleoylethanolamide (oleic acid derivative)	Agonist [217]
	Extracts from Chinese sumac (*Rhus chinensis* Mill.)	Increase PPAR α expression [218]
	Bioactive peptides from corn	Increased expression [219]
	*Lactobacillus kefiri* DH5 *	Upregulation [220]
	*Lactobacillus fermentum* CQPC06 *	Increase PPAR α expression [221]
PPAR γ	Quercetin (red wine, tea, cherries, grapes)	Activation [222]
	Abscisic acid (fruits and vegetables)	Activation [223]
	Gallic acid (tea and fruits)	Partial agonist [224]
	Capsaicin (chili peppers)	Agonist [225]
	Genistein (soybeans and legumes)	Decrease PPARγ levels [226]
	Phycocyanin (blue-green algae)	Downregulation [227]
	Kaempferol	Inverse agonist [228]
	Methoxyeugenol (nutmeg and Brazilian red propolis)	Agonist [229]
	Crocin (saffron)	Activation [230]
	Punicic acid (pomegranate)	Activation [231]
	Linoleic acid (sunflower, soybean, corn, and canola oils, nuts and seeds)	Activation [232]
	Phloretin (apples)	Inhibition [233]
	Phloridzin (apples)	Inhibition [233]
	Equol (eggs and dairy)	Activation [234]
	Daidzein (soybean and legumes)	Activation [234]
	Cinnamon	Activation [235]
	*Lactobacillus rhamnosus* JL1 *	Increased expression [236]
	*Lactobacillus fermentum* TKSN04 *	Upregulation [237]
	*Lactobacillus casei* Zhang *	Increased expression [238]
	*Lactobacillus gasseri **	Activation [239]
	Omega 3 fatty acids	Upregulation [240]
	Fish oil	Decreased expression [241]
	Bioactive peptides: Chia seed peptides Egg white peptides Whey peptides Milk peptides	Inhibition [242] Activation [243] Activation [244] Inhibition [245]
	Phenolic compunds: Mulberry Leaf	Inhibition [246]
	*Glycyrrhiza glabra*	Activation [247]
	*Rumex dentatus*	Upregulation [248]
	Pomegranate juice	Activation [249]
	Canola Meal	Downregulation [250]
	Mango Leaf	Upregulation [251]

* Probiotics. Abbreviations: CBD, cannabidiol; COX, cyclooxygenase; EGC, enteric glial cells; NFkB, nuclear factor-κB; iNos, inducible nitric oxide synthase; PEA, palmitoylethanolamide; PPAR, peroxisome proliferator-activated receptors; TNF, tumor necrosis factor; TLR4, toll-like receptor 4; TRPV, transient receptor potential vanilloid.

## Data Availability

Not applicable.

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
