# Peer review of "The Enteric Glia and Its Modulation by the Endocannabinoid System, a New Target for Cannabinoid-Based Nutraceuticals?"

_molecules, 2022, doi:10.3390/molecules27196773_

Round 1

Reviewer 1 Report

In this revew the Authors explore the the role of the enteric glial cells and endocannabonoid system on gastrointestinal (GI)  tract functions and dysfunctions. They highlighted important connections between endocannabinoid system and the enteric glia and also  the effects of natural cannabinoids and cannabinoid-like molecules with potential nutraceutical use in GI dysfunction.

The review is well organized and easily readable. 

Author Response

Dear reviewer,

Thank you very much for your time and positive consideration towards our manuscript. You can find our responses to all reviewers in the attachment.

Kind regards,

Dr Raquel Abalo (on behalf of all coauthors)

Reviewer 2 Report

This review paper, by Lopez-Gomez et al., compiles current information on the enteric nervous system focusing enteric glial cells, the endocannabinoid system, and its potential utility in GI disorders. Considering the growing prevalence of cannabis and the availability of cannabis-based products along with the general public’s interest on their medical utility, this is a very timely review on an important human health-relevant topic. The contents are appropriate, comprehensive and well-organized with a good number of citations. In general, the text reads well, but there are typos, overly complex/long sentences and rather atypical English expressions that make it difficult for the readers to fully understand the content. The following minor concerns and corrections should be addressed before publication.

Minor:

Line 143-146: The sentence is too complex and not specific enough to have a clear understanding. Does it describe calcium imaging with optogenetic stimulation? 

Line 333-336: The information regarding FAAH-2 and FAAH-1 is misleading. It is not correct nor supported by the citations.

Line 520: The authors should remind the authors that Hemp is defined to have less than 0.3 % Δ9-tetrahydrocannabinol, and it is a very minor ingredient in Hemp. 

Corrections suggested:

Line 20: 

 the enteric glial cells (EGCs) have been comparatively neglected. ïƒ  the enteric glial cells (EGCs) have received less attention.

Line 40:

 the autonomic nervous system (ANS) extrinsic innervation ïƒ  extrinsic innervation of the autonomic nervous system (ANS)

Line 49:

i.e., cannabinoid hyperemesis [19,20] small bowel intussusception, [21] ïƒ  i.e., cannabinoid hyperemesis [19,20] and small bowel intussusception [21]

Line 50:

ECs ïƒ  ECBs? Endocannabinoids?

Line 51:

the relationship between ECs and EGCs and how nutraceuticals may act through it has not been established yet. ïƒ the relationship between ECs and EGCs and the mechanism by which nutraceuticals may act through it have not been established yet.

Line 86:

acetylcholine - ACh (excitatory neurons) ïƒ  acetylcholine (ACh)(excitatory neurons)

Line 170:

several molecules produced by EGCs have emerged as involved in the intestinal barrier formation ïƒ  several molecules produced by EGCs have emerged to be involved in the intestinal barrier formation

Line 232-236:

This sentence is too long and needs to be rewritten.

Line 263-264:

the neuroplastic effects in the PAG area are unlikely due to cause a direct inhibition of central glia. Thus, EGCs have a pro-inflammatory and pro-nociceptive role during colitis, suggested to occur through the selective recruitment of mast cells and activated macrophages…. ïƒ  the neuroplastic effects in the PAG area are unlikely due to a direct inhibition of central glia. Thus, EGCs have a pro-inflammatory and pro-nociceptive role during colitis, which is suggested to occur through the selective recruitment of mast cells and activated macrophages….

Line 275-: Table 2 is too bulky. Contents in the “Changes on EGCs” column can be further summarized.

Line 486: 

PEA exerts dose-dependent anti-inflammatory effect mouse models of UC and in human biopsies. ïƒ  PEA exerts dose-dependent anti-inflammatory effect on mouse models of UC and in human biopsies.

Line 553: 4. Conclusions

Author Response

(The authors gave the same response as above.)

Reviewer 3 Report

1.       Line 37. Please define the term “homeostasis” right here.

2.       Line 52-55. Please revise this paragraph. Avoid the use of “we will” because you did it already. Also, please provide the applicability of the work in this section.

3.       Line 65-79. References are needed.

4.       References are needed for the data in Table 1.

5.       Section 2.2.1. It seems like Ca2+ is only the ion influencing the intercellular communication. How about other ions or other activators?

6.       Line 166-167. “One of the important proteins involved in the intestinal barrier support is glucagon-like peptide 2 (GLP-2).” GLP-2 is protein or peptide?

7.       Line 296. Please provide the molecular structure and chemistry of “canabinoid”

8.       Table 3. How about other neutraceuticals and functional ingredients which are recent trend? Bioactive peptides, essential fatty acids, phenolics….

9.       The effect of drug on the enteric glia and its modulation by the endocannabinoid 2 system should also be included.

10.   The term “neutraceutical” should be defined in the manuscript.

Author Response

(The authors gave the same response as above.)

Round 2

Reviewer 3 Report

All points raised by reviewers were carefully addressed and answered point-by-point. So, it can be accepted.